# Single-Cell Transcriptomics Links Loss of Human Pancreatic β-Cell Identity to ER Stress

**DOI:** 10.3390/cells10123585

**Published:** 2021-12-19

**Authors:** Nathalie Groen, Floris Leenders, Ahmed Mahfouz, Amadeo Munoz-Garcia, Mauro J. Muraro, Natascha de Graaf, Ton. J. Rabelink, Rob Hoeben, Alexander van Oudenaarden, Arnaud Zaldumbide, Marcel J. T. Reinders, Eelco J. P. de Koning, Françoise Carlotti

**Affiliations:** 1Department of Internal Medicine, Leiden University Medical Center, 2333 ZA Leiden, The Netherlands; n.groen@hubrecht.nl (N.G.); f.leenders@lumc.nl (F.L.); a.munoz_garcia@lumc.nl (A.M.-G.); n.de_graaf@lumc.nl (N.d.G.); a.j.rabelink@lumc.nl (T.J.R.); e.dekoning@lumc.nl (E.J.P.d.K.); 2Leiden Computational Biology Center, Leiden University Medical Center, 2333 ZA Leiden, The Netherlands; a.mahfouz@lumc.nl (A.M.); m.j.t.reinders@lumc.nl (M.J.T.R.); 3Delft Bioinformatics Lab, Delft University of Technology, 2628 XE Delft, The Netherlands; 4Department of Human Genetics, Leiden University Medical Center, 2333 ZA Leiden, The Netherlands; 5Hubrecht Institute, KNAW (Royal Netherlands Academy of Arts and Sciences), 3584 CT Utrecht, The Netherlands; m.muraro@scdiscoveries.com (M.J.M.); a.vanoudenaarden@hubrecht.eu (A.v.O.); 6Molecular Cancer Research, University Medical Center Utrecht, 3584 CT Utrecht, The Netherlands; 7Department of Cell and Chemical Biology, Leiden University Medical Center, 2333 ZA Leiden, The Netherlands; r.c.hoeben@lumc.nl (R.H.); a.zaldumbide@lumc.nl (A.Z.)

**Keywords:** human pancreatic islets, β-cells, ER stress, islet integrity, single-cell RNAseq, type 2 diabetes

## Abstract

The maintenance of pancreatic islet architecture is crucial for proper β-cell function. We previously reported that disruption of human islet integrity could result in altered β-cell identity. Here we combine β-cell lineage tracing and single-cell transcriptomics to investigate the mechanisms underlying this process in primary human islet cells. Using drug-induced ER stress and cytoskeleton modification models, we demonstrate that altering the islet structure triggers an unfolding protein response that causes the downregulation of β-cell maturity genes. Collectively, our findings illustrate the close relationship between endoplasmic reticulum homeostasis and β-cell phenotype, and strengthen the concept of altered β-cell identity as a mechanism underlying the loss of functional β-cell mass.

## 1. Introduction

Type 2 diabetes mellitus (T2D) is a complex metabolic disorder caused by failure of pancreatic β-cells in the presence of peripheral insulin resistance. Studies in mouse models led to the concept of β-cell dedifferentiation as one of the mechanisms of β-cell dysfunction in diabetes [1,2]. While most studies on β-cell dedifferentiation rely on the forced activation or deletion of β-cell transcription factors in murine models, we and others found evidence of alterations in β-cell identity in islets from patients diagnosed with type 2 diabetes, such as an increased frequency of polyhormonal cells, a reduced expression of key β-cell transcription factors (e.g., MAFA, PDX1) as well as an increased proportion of degranulated islet cells [3,4,5] β-cell function (assessed on isolated islets) was found to be inversely correlated with the higher proportion of hormone-negative cells in islets from T2D individuals [5]. Similarly, the higher frequency of islet cells presenting an altered identity was correlated with the presence of islet amyloid deposits, which are associated with a reduced functional β-cell mass in type 2 diabetes [6]. Collectively these findings point towards a causative link between β-cell identity change and reduced functional β-cell mass.

In vitro models for β-cell identity changes in primary human islets are limited. We recently reported the downregulation of β-cell maturity markers, particularly MAFA, in drug-induced diabetes [7]. In addition, we previously described a model, in which disruption of primary human islet integrity triggered severe phenotypic alterations in β-cells including the conversion of part of insulin-producing β-cells into glucagon-positive α-cells [8]. Importantly, this process could be prevented by inactivation of ARX, leading to preserved β-cell identity and function.

In recent years, single-cell RNA sequencing (scRNAseq) has emerged as a powerful tool to study complex cell populations including the human pancreas [9,10,11,12,13]. Here, we combine lentivirus-mediated lineage tracing and single-cell transcriptomics to investigate the molecular mechanisms underlying the response of primary human β-cell to cellular stress.

## 2. Materials and Methods

### 2.1. Cell Culture

Islets were isolated from donor pancreas allocated (after anonymisation) by Eurotransplant for the clinical islet transplantation program of our institute (Leiden University Medical Center). Islets were used for research only if they could not be used for clinical purposes, and if research consent was present, according to Dutch national laws. Human islets preparations from 35 non-diabetic donors were used for this study (Appendix A) [14]. Islets with a purity of at least 75% were cultured as previously described [8].

EndoC-βH1 cells [15] and EndoC-βH3 cells [16] were obtained from Univercell Biosolutions (Toulouse, France). Cells were seeded in ECM-fibronectin pre-coated plates containing low glucose DMEM supplemented with 5.5 μg/mL human transferrin, 10 mM nicotinamide, 6.7 ng/mL selenit, 2% BSA fraction V, 50 μM β-mercaptoethanol and pen/strep. EndoC-βH3 cells were treated for 21 days with 1 μM tamoxifen for maturation [16].

### 2.2. Experimental Conditions

#### 2.2.1. Islet Integrity Disruption Model

Two distinct experimental setups were performed to investigate the β-cell response to stress.

Experimental setup 1: Islets were dispersed into single cells by incubation with 0.025% trypsin (Gibco) supplemented with 10 mg/mL DNase (Pulmozyme, Genentech, San Francisco, CA, USA) for 6–7 min and filtered through a 70 μm cell-strainer. For β-cell lineage tracing, the two lentiviral vectors pTrip-RIP405Cre-ERT2-ΔU3 (RIP-CreERT2) and pTrip–loxP-NEO-STOP-loxP-eGFP-ΔU3 (CMVstopGFP) were used, kindly provided by P. Ravassard [17]. Dispersed cells were transduced overnight with a 1:1 mixture of the two lentiviruses in regular CMRL medium containing 8 μg/mL polybrene. In the morning the medium was refreshed, and 4-hydroxy-tamoxifen (Sigma-Aldrich, St. Louis, MO, USA) was added to a final concentration of 1 μmol/L in the evening to activate RIP-CreERT2 and induce GFP expression in β-cells specifically. After overnight incubation, the medium was refreshed and all cells were seeded onto 2% *w*/*v* agarose microwell chips containing 2865 microwells/chip with a diameter of 200 μm/microwell [18,19], resulting in cellular aggregates of ~1000 islet cells (Appendix A).

Experimental setup 2: Intact islets were transduced overnight with the two lentiviral vectors followed by tamoxifen exposure, as described above. After 3–4 days, transduced islets were dispersed and prepared for scRNAseq (Appendix A).

#### 2.2.2. Other Experimental Models

Human islets or EndoC-βH1 cells were treated with either 0.1 µM thapsigargin (TG, Bio-Connect, Huissen, the Netherlands) for 24 h or 1 µM TG for 5 h. Read-outs were performed 24 h after the start of the treatment with TG, unless stated otherwise in the figure legends.

Human islets and EndoC-βH3 cells were treated with 0.1 µM jasplakinolide (JP, Biovision, San Jose, CA, USA) for 72 h or 1 µM JP for 24 h.

### 2.3. Cell Sorting and scRNAseq

Islets and aggregates cultured as outlined in experimental setup 1 and 2 were briefly washed in PBS and dispersed into single cells. Cells were sorted using a FACS Jazz or FACS Aria III (BD Biosciences, Franklin Lakes, NJ, USA). Live single cells were selected based on DAPI exclusion and FSC (Appendix A), sorted into 384-well hard shell plates (BioRad, Hercules, CA, USA) and further processed with SORTseq (SOrting and Robot-assisted Transcriptome sequencing) as previously described [9].

Islets from donors 1, 2 and 3 were used in experimental setup 1, and donors 4, 5 and 6 for setup 2. Of note, donor 1 is has been studied on days 3 and 7 while the other donors have been processed on days 2, 5 and 7 for practical reasons (limited accessibility to the FACSsorting facility). In addition, scRNAseq data from intact islets from donor 1 and 2 were combined with our previously generated data from the same donors (and using the same SORTseq protocol) (donor D25 and D28, respectively—GEO: GSE85241 [9]) to increase cell number from intact islets as reference.

A detailed bioinformatics analysis is presented in Appendix B. Briefly, the raw sequencing data were mapped to the human reference transcriptome. The resulting count tables were filtered to remove low-quality cells. Dimensionality reduction of the data was performed using tSNE [20] and cells were clustered using graph-based clustering using Seurat [21] to identify all the pancreatic cell types. Differential expression analysis was performed using the Wilcoxon rank-sum test, and *p*-values were adjusted with Benjamini-Hochberg correction. Significant genes (Padj < 0.05) were subjected to pathway analysis using Ingenuity Pathway Analysis (www.ingenuity.com, accessed on 5 July 2018).

The transcriptional profiles of dispersed β-cells were compared to scRNAseq data of islets from individuals with T2D [10] using the rank-rank hypergeometric overlap (RRHO) algorithm [22]. We used pseudo-temporal ordering of lineage traced β-cells in the context of canonical α- and β-cells. Of note, potential changes due to culture time were taken into account in the analysis by comparing data with control cells sorted at the same time point (data not shown).

### 2.4. Immunofluorescence Microscopy and Flowcytometry

For whole mount immunofluorescence microscopy, formalin-fixed islets were permeabilised using 0.3% triton-X for 1 h and blocked using goat serum for 1 h. Primary and secondary antibodies were sequentially incubated for 24 h and 12 h respectively with occasional shaking. After counterstaining with Hoechst (BD Biosciences, Franklin Lakes, NJ, USA), samples were mounted using DABCO-glycerol on microscopy slides and confocal imaging on the whole islets was conducted using SP8 WLL (Leica, Wetzlar, Germany).

For flowcytometry, islets were dispersed prior to formalin-fixation and permeabilisation with 0.1% triton-X. After blocking with 2% goat serum for 30 min, single cells were sequentially incubated with primary and secondary antibodies for 20–30 min. Cells were analyzed using an LSRFortessa (BD Biosciences, Franklin Lakes, NJ, USA).

Primary antibodies against C-peptide (Developmental Studies Hybridoma Bank, GN-ID4), glucagon (Abcam, ab92517) and GFP (Aves Labs, GFP-1020) were used. Secondary antibodies were Alexa Fluor 488-, 568- and 647 anti-rat, rabbit or chicken were used when appropriate and incubated together with the nuclear counterstain Hoechst.

### 2.5. Western Blot

Islets (~3000 IEQ per sample) and aggregates were washed with cold PBS and lysed using Laemmli sample buffer (60 mmol/l Tris pH 6.8, 10% Glycerol, 1% SDS, 0.001% blue Bromophenol and 5% β-mercaptoethanol) by passing 10 times through a 25-gauge needle and boiling for 10 min at 95 °C. Lysates were centrifuged, and 10 μL of supernatant was loaded into a SDS-PAGE gel and tested for immunoblotting with antibodies against pEIF2a (Abcam, ab32157), EIF2a (Abcam, ab169528), MAFA (Bethyl, A700-067) and GAPDH. Horseradish peroxidase–conjugated anti-rabbit or anti-mouse was used as secondary antibody, and enhanced chemiluminescent substrate was used to develop the signal. The bands were quantified using the ImageLab software (BioRad, Hercules, CA, USA).

### 2.6. RNA/qPCR

Human islets were washed in PBS and total RNA was extracted using the (Micro) RNeasy kit (Qiagen, Hilden, Germany) according to the manufacturer’s instructions. Total RNA was reverse transcribed using M-MLV reverse transcriptase (Invitrogen, Waltham, MA, USA) and oligo (dT). Quantitative PCR was performed using a CFX system (Bio-Rad, Hercules, CA, USA). Fold change was calculated using the delta CT method with human β-actin or GAPDH as reference gene. Primers used are listed in Appendix A.

### 2.7. Glucose-Stimulated Insulin Secretion

Approximately 50 IEQ per well were placed in a 96-well transwell plate. Glucose-stimulated insulin secretion was performed as previously described [8]. Insulin secretion was assessed using a human insulin ELISA kit (Mercodia, Uppsala, Sweden) following the manufacturer’s instructions.

### 2.8. Statistical Analysis

All data are expressed as means ± SEM, unless stated otherwise. For analysis of qPCR data, statistical significance of differences between two groups was determined by a paired Student’s *t* test on the delta CT values calculated from the reference gene (β-actin or GAPDH) and the gene in question. A *p*-value below 0.05 was considered statistically significant.

## 3. Results

### 3.1. ER Stress Is Associated with Loss of β-Cell Identity in a Model of Islet Integrity Disruption

Disruption of primary human islet integrity triggers severe phenotypic alterations in β-cells including the conversion of part of insulin-producing β-cells into glucagon-positive α-cells [8]. In order to decipher the dynamic process underlying these events, we combined β-cell-specific lineage tracing [8,17] and scRNAseq on primary human islet cells. Briefly, human islets were dispersed into single cells, and transduced with a lentivirus-based β-cell lineage tracing system and all cells were then reaggregated in microwells. From these aggregates, single cells were sorted from the entire live cell population (‘AGG-all’) or enriched for lineage-traced β-cells (‘AGG-GFP+’) at 2, 5 and 7 days post-reaggregation (setup 1 outlined in Appendix A). Single cells from untransduced intact islets (‘ISL-all’) were used as control. In total, 4093 cells from three donors were processed for scRNAseq using setup 1 (Figure 1A,B and Appendix A). In line with our previous findings [8], the detection of GFP+ cells sorted from reaggregated islet cells in the α-cell cluster confirmed the identity change of part of β-cells upon dispersion and reaggregation (Appendix A). The lineage-traced AGG-GFP+ cells detected in the α-cell cluster express all classical α-cell genes (e.g., GCG, ARX, CHGB), and at similar levels as canonical α-cells, while most β-cell genes are found to be downregulated (Appendix AB and Figure 2C). Yet, converted α-cells differ notably from canonical α-cells, displaying higher levels of HLA genes (HLA-A, -B, -C and -E), IFI6 as well as remnant β-cell genes INS and IAPP, and lower levels of α-cell markers ALDH1A, LOXL4 and GC (Appendix A and Appendix A). Altogether these data indicate that converted α-cells are highly similar, but not identical to canonical α-cells.

Next, we aimed at monitoring this dynamic process in the scRNAseq data. We developed an algorithm based on the expression of β- and α-signature genes to define a cell identity score, thereby allowing a pseudo-temporal ordering of the GFP-labelled cells in relation to canonical α- and β-cells (Appendix A, Appendix A). The scores of intact islet cells identify two distinct populations (native β-cells: 0, native α-cells: 1) (Figure 1C and Appendix A). Interestingly, the β-cell peak observed in the AGG-GFP+ cells is off-centred, indicating alterations in β-cell identity. In addition, pseudo-temporal ordering indicates that 3–10% of AGG-GFP+ cells fell into the category of cells with an ‘intermediate’ identity between β- and α-cells (Appendix A), concomitant with a gradual downregulation of typical β-cell markers (PDX1, INS) (Figure 1D and Appendix A). However, no increase in gene expression of endocrine progenitor markers (such as NEUROG3, RFX6, HES1, SOX9) was detected across this pseudo-time analysis (Appendix A), arguing against a typical dedifferentiation process in this model.

The pathway analysis performed on the intermediate cell-specific genes revealed that 30% of the affected genes are coding for components of the ribosome complex and are upregulated in this cell subpopulation. Furthermore, EIF2 signalling and translation-related genes, as well as genes characteristic of the unfolded protein response (UPR) are most affected (Appendix A and Figure 1E). Importantly, the quality of these intermediate cells is similar to other AGG-GFP+ cells (Appendix A). However, the low cell number in one of the donors hampered the statistical power of the comparison. Nonetheless, the combined analysis of intermediate cells across the three donors confirmed the affected pathways found in donor 3 (data not shown).

We specifically evaluated the UPR across identity score pseudo-temporal ordering. Genes involved in all three arms (IRE1α, PERK and ATF6) are mostly affected mostly around the transition between β-cells and intermediate cells (Figure 1F). Genes associated with the IRE1α arm are affected earlier, as shown by increased XBP1 expression just before the intermediate zone. This is followed by genes associated with the PERK arm; ATF4, which is preferentially translated upon EIF2α-induced translation arrest, is affected at gene expression level around the intermediate zone. ATF4 target genes (TRIB3, DDIT3 and PPP1R15A) are upregulated in early intermediate cells, whereas the expression of the target gene ATF3 is increased later. ATF6 expression is lost when an α-cell identity is acquired across identity score, concomitant with the increased expression of the negative regulator of ATF6, WSF1. DDIT3, which can also be a target gene in the ATF6 arm, seems to have a more similar pattern to other PERK-related genes.

Altogether, these data indicate that the β-cells displaying high Endoplasmic Reticulum (ER) stress lose the expression of typical β-cell markers post-disruption of islet integrity, and that the activation of the three arms of the UPR is associated with the transition from mature β-cells to cells displaying severe alterations in identity.

Importantly, in order to validate that the lineage tracing system is faithfully targeting native β-cells at the start of the experiment, we performed a control experiment allowing examination of day 0 (i.e., before dispersion) GFP^+^ cells: we transduced intact human islets (instead of dispersed islet cells as in the first experimental setup); when GFP fluorescence was visible at 3 to 4 days post-transduction, the islets were dispersed and processed for scRNAseq analysis (experimental setup 2; Appendix A–E). As expected, most of the lineage-traced cells were β-cells at this time point, as shown at mRNA level (scRNAseq, Appendix A), and at the protein level, as virtually all analysed ISL-GFP^+^ were C-peptide-positive (Appendix A). Moreover, any potential effect of the lentiviral transduction itself could be ruled out as none of the classical ER stress-responsive genes were found to be differentially expressed in transduced GFP-labelled β-cells, when compared to unlabelled β-cells from intact islets (Appendix A). This is in line with an earlier study where we showed that lentiviral transduction does not alter primary human β-cell function [23].

### 3.2. Stress Signature in the Islet Integrity Disruption Model Displays Similarities with Hallmarks of β-Cell Stress in Type 2 Diabetes

We next examined the cell populations identified as β-cells, and α-cells for comparison. In order to determine the response of these cells to the loss of cell-cell contact and disruption of islet integrity, we compared the gene expression profiles of β-cells from aggregates (AGG-all) to the profiles of canonical β-cells (ISL-all). Islet cell dispersion induces stress-response pathways such as the UPR, protein ubiquitination and NRF2-mediated oxidative stress response, and an attenuation of the translation in a more pronounced manner in β-cells than in α-cells (Figure 2A, Appendix A). At the protein level, we confirmed the increased phosphorylation of the α-subunit of the translation initiation factor eIF2 early after islet cell dispersion (Figure 2B). Moreover, the UPR is quickly activated after dispersion with the immediate upregulation of XBP1s, followed by ATF3 and later CHOP in the whole islet cell population (Figure 2D–F).

In addition, we found iron homeostasis to be affected in β-cells with the downregulation of ferritin subunits FTH1, FTL and FTH1P3 gene expression, and the upregulation of the hypoxia inducible transcription factor EPAS1 and of the transferrin receptor TFRC. Other alterations, although more donor-specific, were observed post-islet integrity disruption. Among these, immune response pathways with overlapping genes related to phagosome maturation and antigen presentation were upregulated in β-cells (donor 3). The significant overrepresentation of this pathway cluster is driven by the upregulation of HLA genes (HLA-A, -B and -C), B2M and FOS upon dispersion. Data from this donor also showed an upregulation of interferon-induced genes (e.g., IFI6, IFIT3, MX1) specifically in β-cells. Finally, the upregulation of glycolytic enzymes in β-cells was observed in donor 1. Mitochondrial function was attenuated in α-cells (donor 1), with the downregulation of subunits of the respiratory complexes.

Prolonged ER stress has been associated with β-cell dysfunction, β-cell death, and the development of both type 1 and type 2 diabetes. We therefore investigated whether the altered gene expression profile in our model of β-cell identity loss displays some similarities with the transcriptional profiles of β-cells from individuals with type 2 diabetes. scRNAseq profiles of islets from patients with type 2 diabetes previously published by Segerstolpe et al. were used for comparison [10]. The dispersion-induced transcriptional profiles from our study significantly overlapped with gene expression profiles in β- and α-cells from subjects with type 2 diabetes (Figure 2C). In β-cells, both up- and down-regulated genes largely correlated with each other, whereas, in α-cells, mostly upregulated genes in intact and non-diabetes cells overlapped.

Furthermore, we specifically checked whether the pathways affected upon dispersion were also altered in cells from type 2 diabetic donors (Figure 2A). Genes involved in translation (EIF2 signalling, mTOR and ‘regulation of eIF4 and P70S6K’ signalling) and stress response (UPR, protein ubiquitination-related pathways) are differentially regulated in β-cells from individuals with type 2 diabetes. Noteworthy, the ER stress and the NRF2-mediated oxidative stress pathways are not detected in the differentially expressed genes in α-cells. Pathways involved in phagosome maturation and iron homeostasis are affected in both cell types. Interestingly, the cell-adhesion-related pathways are also differentially regulated in β-cells from subjects with type 2 diabetes.

### 3.3. ER Stress Leads to β-Cell Dysfunction through Loss of β-Cell Identity

We next asked the question of whether ER stress, caused by islet integrity disruption, is a major driver of the dispersion-induced altered gene expression profile in human β-cells. We subjected primary human islets to treatment with thapsigargin (TG), a typical ER stress inducer, and assessed the effect on key β-cell gene expression and on β-cell function.

Different TG conditions were evaluated. Treatment with 0.1 μM TG for 24 h [24] induced a 3-, 4-, and 6-fold upregulation of the classical ER stress markers XBP1s/XBP1u, ATF3 and CHOP mRNA, respectively (Appendix A). In addition, we evaluated the effect of 1 μM TG for 5 h [25] at 0, 24 and 48 h after the end of the treatment (Appendix A). As expected, the UPR response was transitory, slightly decreasing over time. The ER stress response at 24 h after the end of the 5 h treatment with 1 μM TG was similar to the response seen after the 24 h treatment with 0.1 μM TG (and to the one found in the model of islet integrity disruption (Figure 2D), so these two TG setups were chosen as a model for ER stress in the rest of the study.

We first assessed the effect of ER stress on key β-cell genes. Gene expression of the maturity marker MAFA was strongly decreased upon ER stress induction, both at mRNA (Figure 3A) and protein level (Figure 3C). Similarly, gene expression of the key transcription factor PDX1 and of the regulator of β-cell fate PAX4 was reduced by 30% and 60%, respectively, further indicating the loss of β-cell identity after ER stress induction. In contrast, NKX6.1 and insulin gene expression remained unaffected, hence ruling out that the effects described above may be resulting from β-cell-specific death (Figure 3A).

Next, the effect of ER stress on endocrine progenitor cell markers was assessed. Gene expression of SOX9, HES1 and C-MYC was increased by 55%, 80% and 65% upon TG treatment, respectively, whereas NEUROG3 expression was unaffected (Figure 3B). Of note, the expression of the typical duct marker KRT19 was unaffected, ruling out that the upregulated expression of SOX9 and HES1, which mark not only progenitor cells but also adult duct cells, results from an increased exocrine fraction in the islet preparations. Furthermore, TG treatment (1 μM, 5 h) of the human β-cell line EndoC-βH1 confirmed an upregulation of both SOX9 (90%) and HES1 (60%) (Appendix A). No upregulation of α-cell genes was observed (data not shown).

Glucose-stimulated insulin secretion was decreased after 5 h treatment with 1 μM TG in two of three donors (Figure 3D), suggesting impaired β-cell function due to the occurrence of ER stress.

Overall, these data establish a causal link between ER stress induction and loss of β-cell identity and function, in line with our findings from the islet integrity disruption model.

### 3.4. Altering Actin Cytoskeleton Affects Human β-Cell Identity and Function

Our findings so far indicate that islet integrity disruption, and thus the loss of cell-cell contact, affect β-cell identity via ER stress. To further investigate the importance of islet integrity for the maintenance of β-cell identity and function, we focused on the role of the actin cytoskeleton in β-cells.

Cell adhesion molecules are connected to the actin cytoskeleton and the microtubule network. Actin cytoskeleton remodelling is necessary for insulin granule trafficking and release. Here we hypothesised that actin cytoskeleton remodelling is crucial for the maintenance of β-cell identity and function. To test this hypothesis, we treated human islets with jasplakinolide (JP), a compound with actin polymerizing- and stabilizing capacities, for 24 h. Firstly, JP treatment reduced β-cell function in islets of two out of three donors (Figure 4A). Furthermore, the treatment of primary human islets with JP induces ER stress as seen by a 10-fold upregulation of ATF3 gene expression, a 3-fold upregulation of CHOP gene expression and a mild increase in XBP1s/XBP1u (Figure 4B). In line with our previous observations, ER stress induction is correlated with a clear reduction (50%) in MAFA gene expression (Figure 4C). PDX1, NKX6.1 and insulin gene expression were not significantly altered, ruling out the toxicity of JP on β-cells. No major change in gene expression of SOX9, HES1 and CK19 was observed (data not shown). In addition, these findings were confirmed in a mature human β-cell line (EndoC-βH3), where increased expression of ER stress-related genes is correlated with a clear reduction in MAFA gene expression (Appendix A).

Collectively, these data show that altering the cytoskeleton dynamics induces an ER stress response in β-cells, followed by altered β-cell identity and some degree of impairment in β-cell function.

## 4. Discussion

The current study shows a β-cell adaptation mechanism to ER stress through altered identity and therefore reduced function.

Using a model of loss of islet integrity, we identified a stress signature highly similar to hallmarks of β-cell stress found in type 2 diabetes. As an adaptive response to mild ER stress, the UPR has been shown to modulate β-cell proliferation in response to increased insulin demand [26]. However, more severe ER stress has also been linked to β-cell dysfunction, β-cell death and to the presence of type 1 [27,28,29] and type 2 diabetes [30,31,32,33]. Here we propose that identity change constitutes a novel mechanism of adaptation for β-cells to survive irremediable ER stress, and as an alternative to a more definitive path that would be programmed cell death (Figure 5).

Despite the fact that single-cell transcriptomics analyses are limited to a few donors only, this type of technology allows generating new hypotheses that can then be tested in other types of assays and using more donors/cells. Using this approach, we showed that activation of ER stress is associated with the loss of key transcription factors. Mouse model studies suggested that β-cell failure can be attributed to β-cell dedifferentiation to a progenitor stage and reprogramming to other endocrine cell types [1]. These observations were supported by evidence of altered β-cell identity found in human islets from donors with type 2 diabetes and type 1 diabetes [34], albeit without the detection of a progenitor state [4,5,35,36]. Our findings that MAFA and PDX1 are more susceptible to a stress-induced decrease in mature β-cells are in line with previous findings both in human cells [7] and in mice, where the failure of the adaptive UPR in a diabetic environment is associated with decreased expression of β-cell maturity markers [37]. Furthermore, upon induction of ER stress by TG, we found some evidence of dedifferentiation, with a reactivation of the endocrine progenitor markers SOX9, HES1 and C-MYC in β-cells. This is in line with earlier reports in other in vitro models of β-cell stress [38,39]. In contrast, both islet integrity disruption and JP treatment did not lead to increased expression of progenitor markers. This apparent discrepancy is likely to reflect the effect of additional mechanisms triggered in the latest two conditions.

In the islet integrity disruption model, as in type 2 diabetes, the stress response appears to be more prominent and more persistent in β-cells than in α-cells. β-cells display specifically altered expression of ER stress, UPR, and NRF2-related pathways, while these pathways are not differentially affected in α-cells. Interestingly, α-cells are thought to cope better with ER stress and related apoptosis because of the expression of specific anti-apoptotic proteins [40]. In addition, α-cells have been shown to display higher levels of antioxidant defence mechanisms than β-cells, which make the latter ones more sensitive to the overproduction of reactive oxygen species [41]. These intrinsic differences between α-cells and β-cells may explain a distinct adaptation mechanism to stress from these two cell types.

Our data from the islet integrity disruption model and JP treatment indicate that loss of cell-cell contacts triggers ER stress in β-cells. Importantly, the islet architecture is disrupted in both type 1 and type 2 diabetes, due to the specific destruction of β-cells and the development of amyloid [4], respectively. Cell-cell and cell-ECM interaction, mediated by cadherins and integrins, are critical for the maintenance of β-cell identity and function [42,43]. Recent studies have suggested that cytoskeleton status is crucial for the correct development of β-cells from human pluripotent stem cells [44] and the correct insulin granules movements [45]. Cadherins and integrins participate in adherens junctions where they are connected via catenins and focal adhesions to the actin cytoskeleton and the microtubule network. As the cytoskeleton guides granule movement and exocytosis, dispersion-mediated disruption of the cell-to-cell contacts between the islet cells potentially interferes with β-cell function [46,47]. This may be followed by a disruption of the autoregulatory feedback loop that helps to reinforce the β-cell phenotype [48]. Moreover, independent of their role in the UPR, both PERK and IRE1α has been shown to control actin remodelling and dynamics by serving as a scaffolding protein for the actin crosslinking factor filamin A [49,50]. The actin cytoskeleton is connected to the nuclear envelope and as such extracellular signals can be rapidly transmitted and induce structural changes in the nucleus [51,52]. Altogether, this series of events may explain the altered identity of β-cells. Furthermore, detachment from the extracellular matrix has been associated with elevated ROS levels in epithelial cells. ROS constitute signalling molecules sensing environmental cues [53,54]. Elevated ROS levels have been associated with reduced activity of key transcription factors PDX1 and MAFA in β-cells [55]. As found in our scRNAseq data, oxidative stress and ER stress responses are intertwined, and both processes are known to alter β-cell function. Oxidative stress can induce protein misfolding by disturbing the redox state in the ER, while PERK activation in response to protein misfolding also activates the master regulator NRF2 and the oxidative stress response [56].

In conclusion, we present a mechanism of adaptation of primary human β-cells to cellular stress through the loss of β-cell maturity. This mechanism may play a role in the loss of functional β-cell mass that is associated with the onset and the development of diabetes, and therefore may allow new therapeutic opportunities. First of all, alleviating ER stress may not only help to promote β-cell survival, but also to preserve β-cell identity and therefore function. Additionally, if we assume that cellular plasticity is a reversible process, during a specific time window at least, novel (endogenous) β-cell regeneration therapies could be envisaged for diabetes.

## Figures and Tables

**Figure 1 cells-10-03585-f001:**
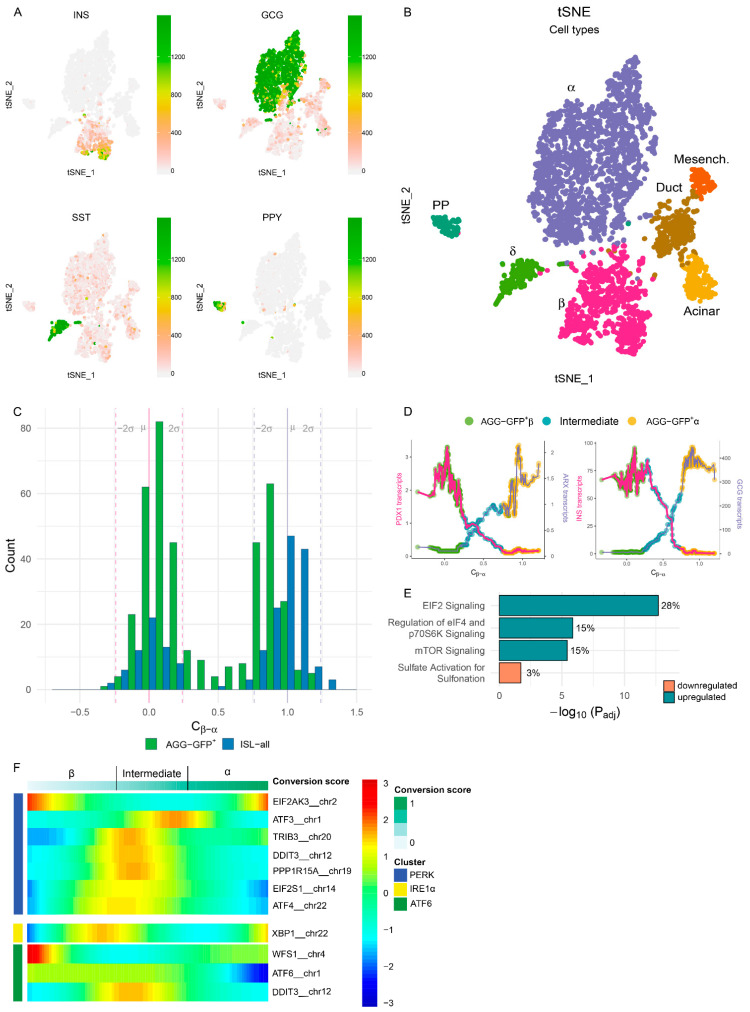
scRNAseq and pseudo-temporal ordering of β-cells show that loss of identity is associated with the upregulation of stress markers. (**A**) In total 4093 cells from three donors were processed for scRNAseq using setup 1 (Appendix A). The obtained clusters were assigned to the pancreatic cell types based on their transcriptional profiles illustrated by the expression levels of representative markers. The colour bar represents the transcript counts of typical endocrine markers. Transcript counts are in linear scale. (**B**) Projection of all sequenced single cells from three donors (D1–3) on a t-SNE map. Colours represent cell type clusters. Cell type numbers are outlined in Appendix A. (**C**) In order to monitor the conversion process in the scRNAseq data, we devised an algorithm to define a cell identity score. Cell identity scores of both canonical (Islet all ‘ISL-all’—blue bars) and AGG-GFP+ (Aggregates GFP+ green bars) β- and α-cells are represented in a histogram (donor 3). The identity score of canonical β-cells is centred around zero and of canonical α-cells is centred around one. The mean conversion scores of both canonical β- and α-cells (μ) and two standard deviations from the mean (±2σ) are indicated by the dashed lines. These population statistics were used to identify ‘intermediate cells’ with a cell identity score in-between both populations. (**D**) Concomitant with pseudo-temporal ordering using the identity score, the expression of PDX1 vs. ARX (left) and insulin vs. glucagon (right) are gradually inversed in AGG-GFP+ cells (donor 3). The dots represent the moving average of the expression values across identity score, coloured by conversion stage (non-converted β-cells, intermediate and converted α-cells). (**E**) Pathway analysis performed on the intermediate cell-specific genes for which differentially expressed genes in intermediate cells (based on both up- and downregulated genes in intermediate cells vs. AGG-GFP+β and AGG-GFP+α cells) were used. Pathways and direction of regulation affected in intermediate cells are shown along with the fraction of genes in that pathway of the total affected genes (donor 3). (**F**) Scaled expression of Unfolded Protein Response (UPR) genes across cell identity score from β- to α-cells (donor 3). The gene-clusters (rows) represent the three UPR pathways.

**Figure 2 cells-10-03585-f002:**
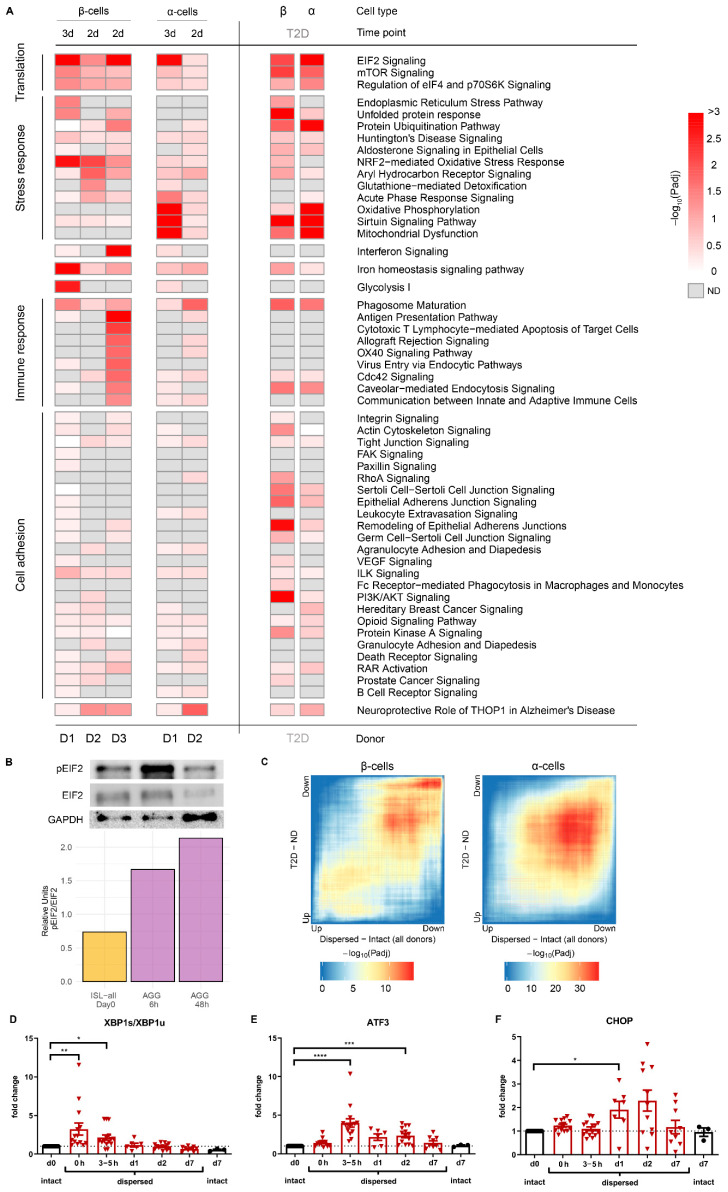
Stress signature displays similarities to hallmarks of β-cell stress in type 2 diabetes (T2D). (**A**) The response of β- and α-cells in the islet integrity disruption model was further characterised and compared to the transcriptional profiles of islet cells from T2D individuals. Pathways affected upon dispersion in β-cells across different donors and time points in comparison to the pathways affected by T2D. Pathway categories were assigned to pathways with similar genes responsible for the overrepresentation of the pathway, and such redundant pathways were grouped and displayed together. The columns represent the β-cells and α-cells from the different donors and timepoints post-dispersion or type 2 diabetes vs ND. For representation purposes, the -log10 (*p*-values) were capped at 3. The timepoint of sorting is annotated above and the donor is annotated below in the figure. (**B**) The ER stress response upon islet integrity disruption is evidenced by the increased ratio of protein levels of phosphorylated-eIF2α (p-eIF2 α) and eIF2α, shown along with the housekeeping protein GAPDH (donor 9). (**C**) The dispersion-induced transcriptional profiles significantly overlap with gene expression profiles in β- and α-cells from subjects with type 2 diabetes. Rank-Rank Hypergeometric Overlap map between dispersed cell profiles (x-axis) and type 2 diabetes cell profiles (y-axis). The colour codes the -log10 transformed hypergeometric *p*-value corrected for multiple testing and shows the strength of the overlap. (**D**–**F**) Dispersion of isolated human islets leads to increased mRNA expression of the ER stress-related genes XBP1s/XBP1u, ATF3 and CHOP as measured by qPCR. Data are presented as means ± SEM of fold change over intact islets at day 0 (t = d0). *n* = 3–16 donors; numbers depicted in the bars indicate the number of donors per time point. * *p* < 0.05, ** *p* < 0.01, *** *p* < 0.0005, **** *p* < 0.0001 vs. intact islets at day 0 (t = d0).

**Figure 3 cells-10-03585-f003:**
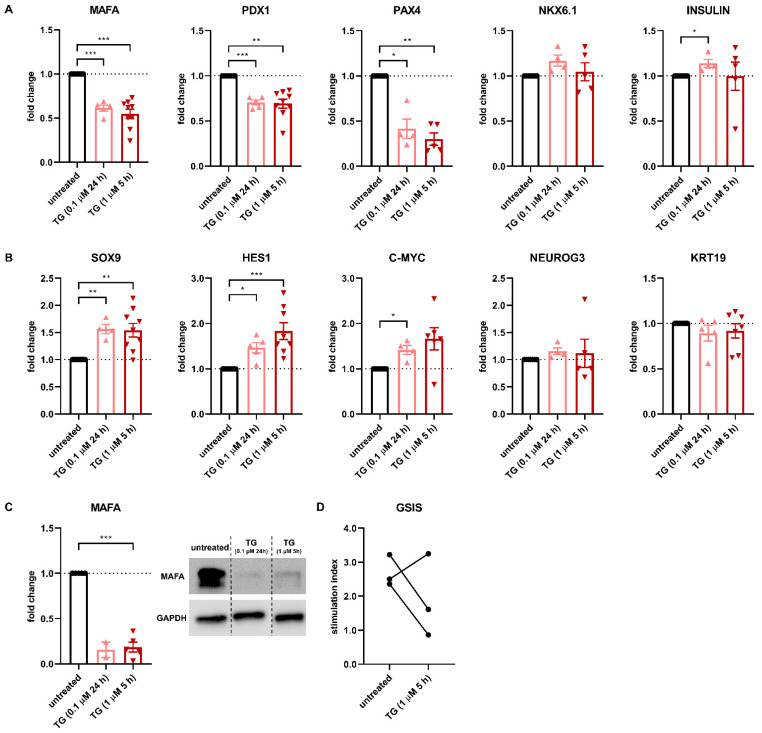
ER stress leads to β-cell dysfunction through loss of β-cell identity. (**A**) To evaluate the effect of ER stress on β-cell identity, islets were treated with the ER stress-inducing compound thapsigargin (TG). Treatment of isolated human islets with 0.1 μM TG for 24 h and 1 μM TG for 5 h leads to decreased gene expression levels of the β-cell-specific genes MAFA, PDX1 and PAX4 as measured by qPCR. (**B**) ER stress induction in human islets by treatment with 0.1 μM TG for 24 h and 1 μM TG for 5 h leads to increased gene expression levels of the endocrine progenitor genes SOX9, HES1 and C-MYC as measured by qPCR. mRNA expression levels of NEUROG3 and KRT19 were unchanged upon TG treatment. (**C**) Levels of β-cell-specific protein MAFA are decreased in islets upon treatment with 0.1 μM TG for 24 h and 1 μM TG for 5 h, as assessed by Western blot. (**D**) TG-induced ER stress in human islets leads to decreased glucose-stimulated insulin secretion in two of three donors. Data are presented as means ± SEM of fold change over untreated control islets. *n* = 4–9 donors; each data point represents one donor. * *p* < 0.05, ** *p* < 0.01, *** *p* < 0.0005 vs. untreated control islets.

**Figure 4 cells-10-03585-f004:**
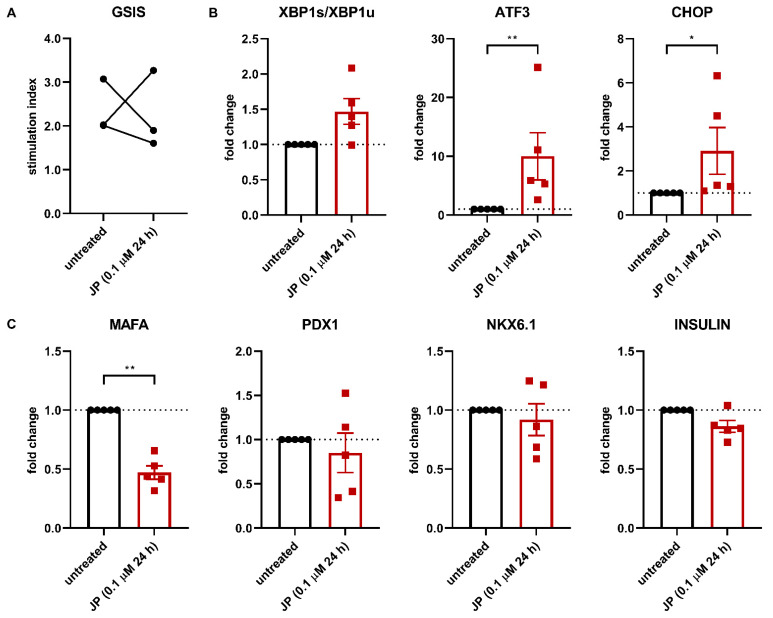
Altering the actin cytoskeleton triggers ER stress and affects β-cell identity and function. (**A**) Islet integrity disruption is further studied by altering the actin cytoskeleton using jasplakinolide (JP). Affecting the actin cytoskeleton in human islets with 0.1 μM JP for 24 h leads to decreased glucose-stimulated insulin secretion in two of three donors. GSIS was performed immediately after 24 h JP treatment. (**B**) Treatment of isolated human islets with 0.1 μM JP for 24 h leads to increased mRNA expression levels of the ER stress marker genes ATF3 and CHOP. (**C**) Treatment of isolated human islets with 0.1 μM JP for 24 h leads to decreased mRNA expression levels of the β-cell-specific gene MAFA but not PDX1, NKX6.1 and insulin as measured by qPCR. Data are presented as means ± SEM of fold change over untreated control islets. *n* = 5 donors; each data point represents one donor. * *p* < 0.05, ** *p* < 0.01 vs. untreated control islets.

**Figure 5 cells-10-03585-f005:**
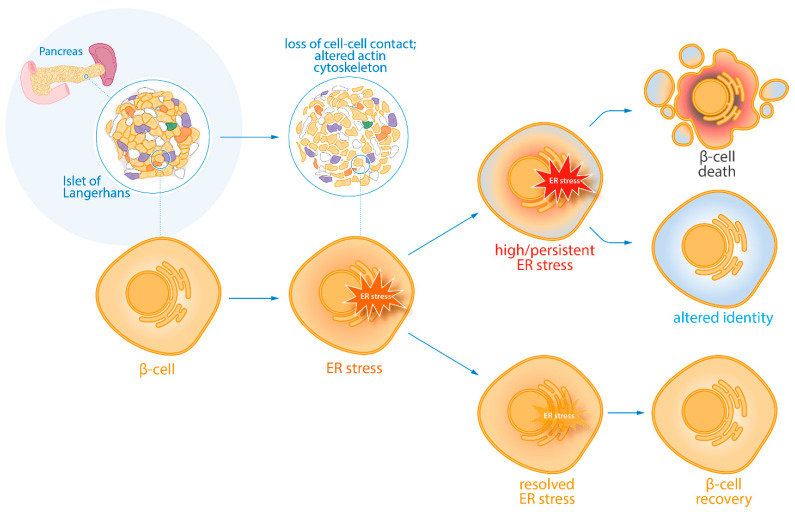
Proposed model. We present a model in which ER stress occurs as a result of the loss of cell-cell contact (in this particular experimental setup) and the subsequent remodelling of the actin cytoskeleton. When ER stress is resolved, the β-cell can fully recover. However, high or persistent ER stress can lead to either β-cell death or altered β-cell identity, thereby leading to a reduced functional β-cell mass. Overall, we propose β-cell identity changes as a cell-intrinsic mechanism to survive irremediable cellular stress. This adaptation mechanism may contribute to the development of diabetes.

## Data Availability

The original contributions presented in the study are included in the article/Appendix A. Further inquiries can be directed to the corresponding author.

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
