# Peer review of "Single-Cell Transcriptomics Links Loss of Human Pancreatic β-Cell Identity to ER Stress"

_cells, 2021, doi:10.3390/cells10123585_

Round 1

Reviewer 1 Report

In the current manuscript, the authors investigated the mechanisms underlying the altered beta cell identity in a model of islet integrity disruption. ScRNAseq combined with lentivirus-mediated lineage tracing were used; data were validated in human islets, EndoC-bH1 and EndoC-bH3 exposed to thapsigargin (TG) or jasplakinolide (JP). The authors found that beta cell adaptation to ER stress is characterized by alteration of beta cell identity, associated to compromised beta cell function. The study is of interest to the field; however, some concerns come up while reading the manuscript.

Major comments:

Experimental design, data analysis and presentation.

1) It is not clear whether GFP+ cells were allowed to form aggregates with all the other islet cells, or they were separated to form specific GFP aggregates (AGG-GFP). If GFP+ cells formed aggregates with all the other cells, it is not necessary to distinguish “AGG-all” and “AGG-GFP” in Figure A1A, it is sufficient just mark the GFP+ cells in the AGG. On the contrary, if GFP+ cells were aggregated separately, the possible interference in the gene expression deriving from the loss of islet cell cross-talk should be considered.

2) Pathway analysis of the intermediate cell-specific genes was performed, and data provided from donor 3 (Figure 1E and F, table A4), this is not enough to generalize the observation; are the data confirmed in the other donors?

3) Does the culture affect the islet cell gene expression? Has this issue been considered?

4) Figure A1A does not well represent the experimental design described in the text. Reaggregation of transduced islet cells start at day 0 (although it proceeds during the culture period), therefore it seems more appropriate to draw the “cell aggregate” at day 0.

5) Figure A3D shows that Donor 1 has been studied at day 3 and 7, instead of day 2, 5 and 7; this does not affect the analysis and the data presented; however, it should be made clear in methods.

6) The authors state “we confirmed the increased phosphorylation of the α-subunit of the translation initiation factor eIF2 within 2 days post-islet cell dispersion (Figure 2B)”; however, data in Figure 2B are questionable since the amount of loaded protein for the AGG sample is higher than those for ISL-all and AGG 6h, as shown by the density of the GAPDH band. A normalization of the data through the amount of protein loaded should be done.

4) References. Some statements are not quoted properly. Examples can be found in references 14, 15, 16, 17, 18, 19 and 20, this makes difficult for the reviewer verify what authors state. Appropriate citation of references should be carefully checked.

Minor comments:

1) The title of the section 3.1 is not appropriate since beta cell function has not been evaluated. 

2) Legend of Figure 3C is not complete, the description of NEUROG3 and KRT19 expression is missed.

3) The way to label the donor islet preparations may be confounding. For the experimental setup 1 donors were indicated as D1, D2, D3; similar labels (donor 1, donor 2, donor 3) have been used for the functional studies with islets exposed to TG or JP. Are donor 1, donor 2 and donor 3 the same as D1, D2 and D3? In addition, were donor 1, donor 2 and donor 3 of the JP experiments the same as for the TG experiments?

4) In Appendix A, references to Figures do not seem to be correct.

Reviewer 2 Report

Very thorough, well written and elegant study that demonstrate/validate at the single cell transcriptomics that disruption of human islets, by simple disaggregation, results in ER-stress and loss/switch of beta cell identity, mimicking type 2 islets. They also found that collapse of the cytoskeleton also leads to ER-stress and dedifferentiation. The authors develop their own algorithm to compare and contrast the various genomic signatures. 

Author Response

Thank you for your positive comments.

Reviewer 3 Report

To get my comments, please see, file attached . 

Round 2

Reviewer 1 Report

Some concerns raised by this reviewer have been addressed; however, point 2) in major comments has not been addressed at all. Regarding it this reviewer remains of the opinion that observations obtained from only one donor can not be generalized, even if results were confirmed by the analysis of combined data from donor 3 and donor 1 (as shown in the rebuttal), also because, the number of intermediate cells from donor 1 was low, as stated by the authors.

Author Response

Comments and Suggestions for Authors:

Some concerns raised by this reviewer have been addressed; however, point 2) in major comments has not been addressed at all. Regarding it this reviewer remains of the opinion that observations obtained from only one donor can not be generalized, even if results were confirmed by the analysis of combined data from donor 3 and donor 1 (as shown in the rebuttal), also because, the number of intermediate cells from donor 1 was low, as stated by the authors.

We agree with the reviewer that observations made in a small number of donors is a limitation of this analysis (a part was added to the Discussion in the previous revision). We now also added a line also in the results section to clarify the technical difficulties we faced during the study (see line 219).